# Long-Term Oxbow Lake Trophic State under Agricultural Best Management Practices

**Richard E. Lizotte Jr.** [1,*] , **Lindsey M. W. Yasarer** [1] , **Ronald L. Bingner** [1] , **Martin A. Locke** [1] **and Scott S. Knight** [2]

[1] USDA-ARS National Sedimentation Laboratory, Oxford, MS 38655, USA; Lindsey.Yasarer@USDA.gov (L.M.W.Y.); Ron.Bingner@USDA.gov (R.L.B.); Martin.Locke@USDA.gov (M.A.L.)

[2] University of Mississippi Field Station, The University of Mississippi, Abbeville, MS 38601, USA; sknight@olemiss.edu

* Correspondence: Richard.Lizotte@USDA.gov; Tel.: +1-662-232-2956

**Abstract:** A key principle of agricultural best management practices (BMPs) is to improve water quality by reducing agricultural-sourced nutrients and associated eutrophication. Long-term (1998–2016) lake summer trophic state index (TSI) trends of an agricultural watershed with agricultural best management practices (BMPs) were assessed. Structural BMPs included vegetative buffers, conservation tillage, conservation reserve, a constructed wetland, and a sediment retention pond. TSI included Secchi visibility (SD), chlorophyll *a* (Chl), total phosphorus (TP), and total nitrogen (TN). Summer TSI 1977 was >80 in 1998–1999 (hypertrophic) and decreased over the first 10 years to TSI 1977 ≈ 75 (eutrophic). TSI 1977 decrease and changing TSI deviations coincided with vegetative buffers, conservation tillage, and conservation reserve. The TSI(SD) decrease (>90 to <70) coincided with vegetative buffers and TSI(TP) decrease (>90 to <75) coincided primarily with conservation tillage and the sediment retention pond. TSI(Chl) increase (<60 to >70) coincided with conservation tillage and vegetative buffer. Results indicate watershed-wide BMPs can modestly decrease summer trophic state through increased water transparency and decreased TP, but these changes are off-set by increases in chlorophyll *a* to reach a new stable state within a decade. Future research should assess algal nutrient thresholds, internal nutrient loading, and climate change effects.

**Keywords:** eutrophication; vegetated buffers; wetlands; conservation tillage; row-crop agriculture





## 1. Introduction

One of the great global challenges of using modern mechanized agricultural practices over the last century and into the 21st century has been the increase in eutrophication of surface waters within agricultural landscapes [1–4]. With a world population approaching an estimated 7.5 billion people [5], the need for food, fiber, and biofuel increases, with a concomitant increase in the demands on agriculture to meet these needs. Lake, river, and other water quality managers require multiple tools and practices to aid in mitigating agriculturally sourced eutrophication. For these reasons, there is a growing need to provide information on the effectiveness of various tools and practices to improve water quality and lessen eutrophication [1,4,6].

Over the last several decades, algal biomass (often measured as chlorophyll *a*) has been accepted as the most useful and widely utilized basis for determining lake trophic state due to established algal biomass relationships with nutrient loads such as nitrogen (N) and phosphorus (P), potentially leading to dissolved oxygen stress including hypoxia (dissolved oxygen < 2 mg L$^{-1}$) [7]. Concepts of limiting N and P inputs to lakes to mitigate algal blooms through control of non-point sources, including agricultural runoff, have been in existence for more than 40 years [6]. Although the nutrient limitation concept has been successfully applied to deeper lakes (>3 m), difficulties in managing eutrophication and rehabilitation in shallow lakes occur as a result of the surface water algal biomass being significantly influenced by fish and/or zooplankton biomass [1,7–10]. Despite

these difficulties, decreasing nutrient concentrations in lakes through reduction in N and P loads is still viewed as an important first step in mitigating eutrophication and lake rehabilitation [7]. A useful tool for lake eutrophication assessment that incorporates the relationship between nutrients and algal blooms has been the Carlson-type trophic state index (TSI) [7] used in several lake studies over the last two decades [3,4,11–17]. The index is valuable because calculation requires relatively few commonly measured lake water quality variables: Secchi depth visibility, nutrients (nitrogen, phosphorus), and chlorophyll a [3,4,7]. TSI has been utilized for lake management of water use (e.g., drinking water, fish, and wildlife use) [11,12], assist in lake ecoregional nutrient criteria development [13,14], assess the impacts of land-use, such as agriculture, on lake trophic state [3,4,15], and assess lake restoration success or recovery from storm events [16,17].

Agricultural best management practices (BMPs) have been implemented globally as a mechanism for managing soil and water resources while maintaining or improving agricultural production [18–20]. Several of these same practices have the potential to mitigate impacts of agricultural activity on receiving water bodies such as lakes [21,22]. Proper water resource management, such as water quality, can be aided by BMPs that can intercept, mitigate, and process agricultural contaminants such as nutrients [18,20]. However, there is limited information on the effectiveness of BMPs in controlling eutrophication in agricultural watershed lakes [21–24]. Within the US, Federal agencies such as the US Department of Agriculture (USDA), work with landowners on a voluntary basis to implement BMPs for the primary purpose of reducing agricultural pollutant loads to surface waters such as suspended sediment, nutrients (N and P), and pesticides [23,25–27]. One of the underlying assumptions in reducing these agricultural pollutant loads is that receiving water bodies would be less impacted by eutrophication, potentially leading to improved water quality and ecosystem services [25]. Objectives of the current study are to assess the following: (1) what is the long-term (19-year) summer trophic state of an agriculturally influenced oxbow lake, Beasley Lake? (2) Has summer lake trophic state changed during the long-term assessment period? (3) Are there any trends in any changes to summer lake trophic state during the study period? (4) Are these trends in altered summer lake trophic state associated with multiple BMPs implemented in the watershed? (5) Are these associations between altered summer lake trophic state and implemented watershed BMPs indicative of mitigated eutrophication?

## 2. Materials and Methods

### 2.1. Study Site Description

The 625-ha watershed study is within the broader lower Mississippi River alluvial plain and located in Sunflower County, MS (33°24′15″ N, 90°40′05″ W). Beasley Lake is a horseshoe or oxbow lake formed by fluvial activity and isolated from an adjacent river, the Big Sunflower River (Figure 1). The lake is a shallow (<3 m) discontinuous polymictic lake with weak to moderate intermittent thermal and clinograde dissolved oxygen stratification during summer [23].

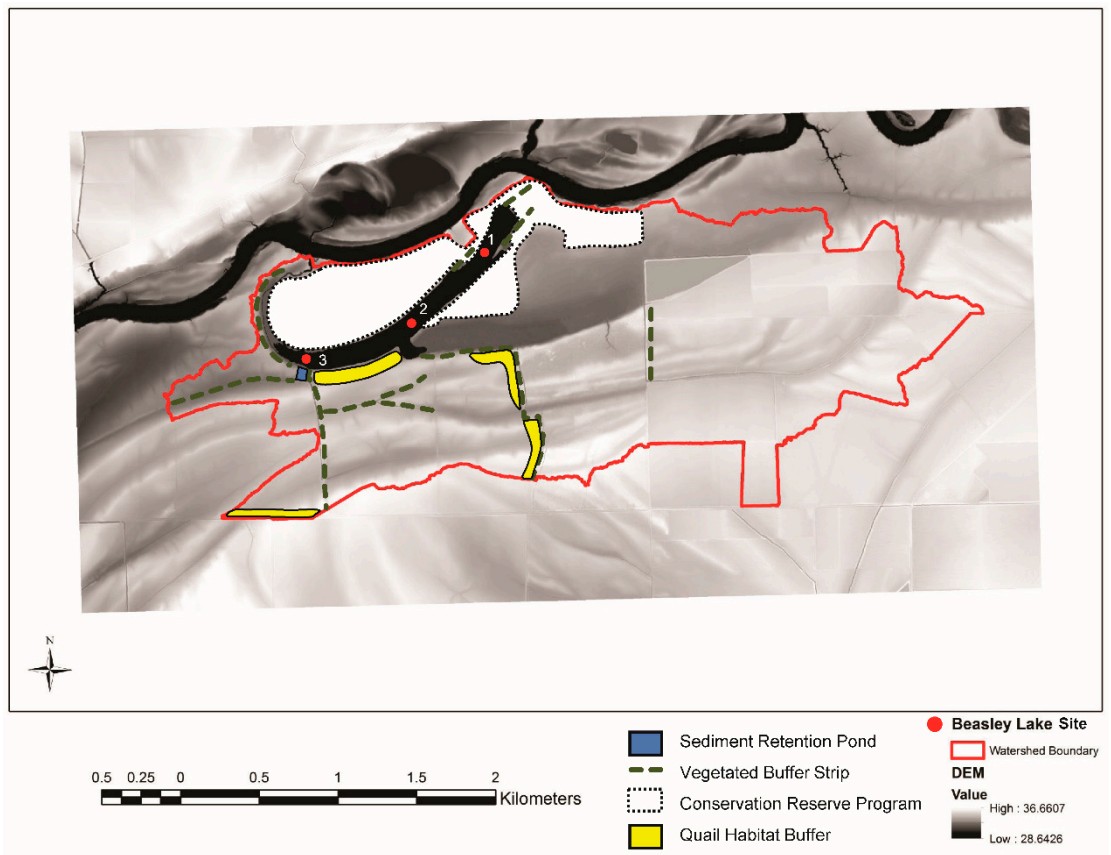

**Figure 1.** Light detection and ranging (LiDAR) remotely sensed digital elevation satellite image of Beasley Lake watershed, Mississippi, with lake surface water sampling sites and locations of best management practices: sediment retention pond, vegetated buffer strips, Conservation Reserve Program, and quail habitat buffer.

Lake morphology is typical for fluvial lakes with Z (average depth) = 2.4 m, $Z_{max}$ = 2.9 m, and surface area >250,000 m$^2$ with an approximate volume of 361,000 m$^3$. The lake has a significant fetch ($L_{max}$ = 1140 m) and narrow width ($La_{max}$ = 95 m) with a moderate shoreline development index ($D_L$ = 2.22) typical of waterbodies in the region. Hydrology of the lake is exhibited by intermittent flow with average hydraulic retention time (HRT) of about three months and a minimum HRT of 5–6 days (Table 1).

**Table 1.** Beasley Lake, Mississippi morphological characteristics.

| Parameter | Abbreviation | Units | Value |
|---|---|---|---|
| Area | A | m$^2$ | 251,202 |
| Volume | V | m$^3$ | 360,917 |
| Maximum length | $L_{max}$ | m | 1140 |
| Maximum width | $La_{max}$ | m | 95 |
| Maximum depth | $Z_{max}$ | m | 2.9 |
| Average depth | Z | m | 2.4 |
| Perimeter | M | m | 4400 |
| Shoreline development index | $D_L$ | unitless | 2.22 |
| Minimum hydraulic retention time | $HRT_{min}$ | days | 5.6 |
| Average hydraulic retention time | HRT | days | 87 |
| Maximum flow rate | $Q_{max}$ | m$^3$ s$^{-1}$ | 2.02 |
| Average flow rate | Q | m$^3$ s$^{-1}$ | 0.0376 |

Land-use within the watershed surrounding the lake consists of 150 ha of non-arable bottomland hardwood riparian wetland and forest and 339 ha of arable land in row-crop production during the entire 19-year study period (1998–2016). Primary crops produced during this period were: cotton (*Gossypium hirsutum* L.), soybeans [*Glycine max* (L.) Merr.], corn (*Zea mays* L.), and sorghum [*Sorghum bicolor* (L.) Moench]. Planting of wheat (*Triticum aestivum* L.) and rice (*Oryza sativa*) occurred only occasionally (in 1998, 2002, and 2012) in limited acreage (<40 ha). Much of the remaining watershed (minus surface water areas) consisted of five measured (as area) independent best management practices implemented from 1995 through 2010. In 1995, the first measured structural edge-of-field BMP, vegetative buffer strips (VBS), was initially installed and encompassed 2.9 ha along the west side of the lake (Figure 1). From 1995–1996, another 1.6 ha of VBS composed of switchgrass (*Panicum virgatum* L.) or fescue (*Festuca arundinacea* Schreb.) were installed [24] and in 2001 an additional 4.6 ha VBS composed of bahiagrass (*Paspalum notatum* Flugge) were installed. Beginning in 2001–2002, the second measured BMP was implemented as conservation tillage management (CT) for cotton and soybean crops across varying watershed area from year-to-year (Figure 2a). Conservation tillage practices are not permanent structural conservation practices but cultural ones that are decided by individual farmers from year to year in the watershed. As such, the areas of conservation tillage are variable. Similarly, fertilizer application decisions were made by farmers and typically included applications added in spring months (March–June) prior to planting a crop, and was usually knifed in. Nitrogen fertilizer, primarily as urea–ammonium–nitrate, was applied every year except 2008–2010. Phosphorus fertilizer was only occasionally applied with the largest application occurring in 2005. Potassium and sulfur were applied frequently from 1998–2005, but only occasionally thereafter (Figure 2b).

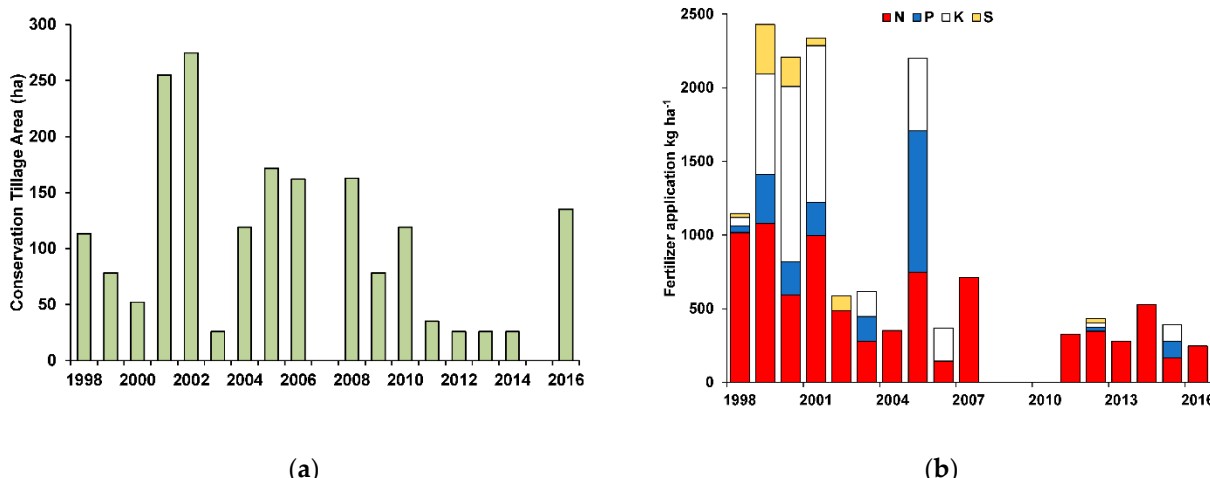

**Figure 2.** Beasley Lake Watershed, Mississippi land management from 1998–2016 as: (**a**) Implemented conservation tillage area (in hectares) and (**b**) Fertilizer application (in kilograms per hectare) as nitrogen (N), phosphorus (P), potassium (K), and sulfur (S).

In 2003, the third measured BMP was implemented within the Conservation Reserve Program (CRP), encompassing 87 ha of arable land removed from row-crop production north of the lake and planted in eastern cottonwood trees (*Populus deltoides* Bartr. Ex. Marsh.), oak trees (*Quercus* sp.), and hickory trees (*Carya* sp.) [24,28] (Figure 1). At this time, about 0.9 ha of VBS as well as a constructed wetland along the north and east shoreline of the lake were subsumed into CRP [29]. In 2006, the fourth measured BMP was installed as vegetative buffer habitat to attract northern bobwhite quail (*Colinus virginianus*) (quail buffer habitat, QB) that encompassed the removal of 9 ha of arable land in the watershed (Figure 1) [23]. In 2010, the fifth and most recent BMP was installed as a 0.99 ha vegetated sediment retention pond (SP) receiving runoff from two drainage ditches draining a total of 125.8 ha south and west of the lake (Figure 1) [30].

### 2.2. In-Situ Water Measurements, Sampling, and Analysis

Surface water quality monitoring and surface water (5 cm depth) samples were collected at three replicate georeferenced sites within Beasley Lake according to Cullum et al. [28]. Briefly, biweekly lake monitoring and 1-L surface water samples were collected at each site from January 1998 to November 2016. In conjunction with surface water sampling at each site, water transparency was measured as in-situ depth of Secchi-disc visibility according to methods described by Wetzel and Likens [31]. Immediately after collection, samples held on ice (4 °C), transported to the USDA-ARS National Sedimentation Laboratory, Oxford, MS, and processed for water quality analysis. Selected water quality parameters analyzed in the laboratory for this study were total phosphorus (TP), total nitrogen (TN), and chlorophyll *a* (Chl). Total N data were not available until 2001.

Laboratory analytical methods used to measure TP, TN, and chlorophyll *a* in water samples are described by Eaton et al. [32]. In brief, TP was determined using the persulfate digestion method, and TN was estimated using the semi-micro Kjeldahl N method with the addition of measured soluble nitrate–N and nitrite-N (as described below). For soluble nitrate–N and nitrite-N analysis, sample water was first filtered through a 45-μm cellulose nitrate filter. Nitrate–N was determined using the cadmium reduction method and nitrite-N was determined using the diazotization method. Following filtration, the filter was removed and placed in acetone solvent to determine extracted Chl using the trichromatic method.

To calculate trophic state, a series of algorithms were used modified from Carlson [33,34] and Kratzer and Brezonik [35]. Trophic state index (TSI) scores were generated according to the US Environmental Protection Agency [36] for each water quality constituent as:

Secchi visibility depth (SD):

$$\text{TSI(SD)} = 60 - [14.41 \times ln(\text{SD}/100)], \tag{1}$$

Total phosphorus (TP):

$$\text{TSI(TP)} = 4.15 + [14.42 \times ln(\text{TP}/1000)], \tag{2}$$

Total nitrogen (TN):

$$\text{TSI(TN)} = 54.45 + (14.43 \times ln\text{TN}), \tag{3}$$

Chlorophyll *a* (Chl):

$$\text{TSI(Chl)} = 30.6 + (9.81 \times ln\text{Chlorophyll } a), \tag{4}$$

Carlson's 1977 TSI is the average of the sum of three individual components: TSI(SD), TSI(TP), and TSI(Chl), whereas Carlson's 1992 TSI is the average of the sum of four individual components: TSI(SD), TSI(TP), TSI(TN), and TSI(Chl). Generated scores can be categorized as: oligotrophic (<40); mesotrophic (40–50); eutrophic (50–70); and hypertrophic (>70) (Figure 3a,b).

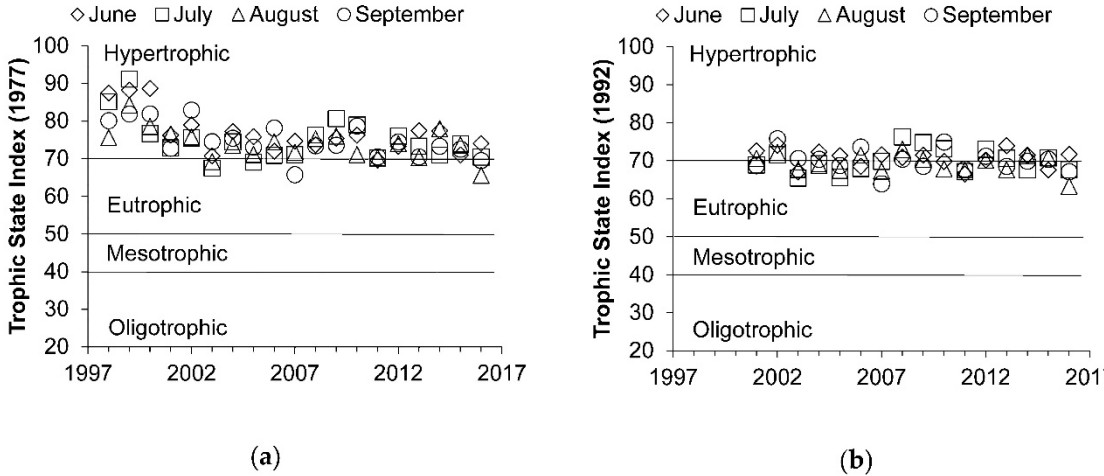

**Figure 3.** Beasley Lake, Mississippi, annual summer (June–September) median trophic state index scores for: (**a**) Carlson-type 1977 [33] from 1998–2016 and (**b**) Carlson-type 1992 [34,35] from 2001–2016.

### 2.3. Trophic State Data Analysis

Summer samples, defined as samples collected from 1 June through 30 September, were selected to determine trophic state during peak algal growing period. A variety of analytical techniques were used to assess changes in the trophic state of Beasley Lake during the study period. Graphical analysis of TSI scores was used according to methods described by Carlson and Havens [37] and Havens and Ji [17]. In brief, deviations in TSI(Chl) scores from TSI(SD), TSI(TP), and TSI(TN) were calculated as differences and then graphed along an x–y axis as follows: TSI(Chl)–TSI(SD) versus TSI(Chl)–TSI(TP) and TSI(Chl)–TSI(SD) versus TSI(Chl)–TSI(TN). Plots are used to infer temporal changes in biotic and abiotic relationships such as non-algal light limitation and/or potential nutrient limitation within Beasley Lake. Because of the frequent non-parametric nature of water quality data [38,39], statistical analyses were non-parametric tests. Trophic State Index scores were analyzed for temporal trends across years using Mann–Kendall Tau Statistic, $\tau$, with two-sided test for assessing increasing or decreasing trends. For TSI scores that indicated significant temporal trends, scores were then assessed for relationships with implementation and integration of multiple BMPs within the watershed. This was conducted using classification and regression tree (CART) analysis utilizing automatic interaction detection with the least squared loss method to determine change points [40,41] when TSI scores (dependent variable) changed in association with BMPs (independent variable). The CART analysis used TSI score data to produce proportional reduction in error (PRE) and improvement values as goodness-of-fit tests that are equivalent to multiple $R^2$ values in parametric multiple linear regressions. Criteria for stopping CART analysis proceeded with a 0.05 split index value and a 0.05 minimum improvement in PRE. Node maximum allowed was set at 21, with a minimum count of 5 allowed in each node. Temporal trend and regression analyses were conducted using SYSTAT statistical package software (SYSTAT v. 13, Systat Software Inc., San Jose, CA, USA).

### 3. Results

#### 3.1. General Water Quality and Trophic State

During the 19-year study period, Beasley Lake summer conditions exhibited a eutrophic to hypertrophic state as determined by limited water clarity, elevated TP and TN concentrations, as well as Chl concentrations frequently exceeding 25 μg L$^{-1}$. Lake Secchi visibility, TN, TP, and Chl values often produced TSI scores > 65, indicating eutrophication (Table 2; Figure 3a,b). Median summer water clarity measured as Secchi visibility over 19 years ranged from 0.36 m in September to 0.47 m in July with most values between 0.10 and 0.60 m. A maximum Secchi visibility depth of 0.93 m was reported in June

2008. As is typical of eutrophic lakes, summer TP and TN concentrations often exceeded 0.05 mg TP $L^{-1}$ and 0.65 mg TN $L^{-1}$, respectively. Median nutrient concentrations ranged from 0.33 to 0.49 mg TP $L^{-1}$ and 1.28 to 1.49 mg TN $L^{-1}$, with the lowest concentrations occurring in July 2016 and August 2007 for TP and TN, respectively (Table 2). Summer lake Chl concentrations ranged widely. However, peak summer productivity often produced concentrations exceeding 100 µg $L^{-1}$. Highest summer Chl concentration, 483 µg $L^{-1}$, occurred in June 1999 and concentrations >200 µg $L^{-1}$ were observed during summer 1999, 2002, and 2003. By comparison, highest summer Chl concentrations from 2004–2006 never exceeded 100 µg/L, whereas from 2007–2016, at least one sample exceeded 100 µg $L^{-1}$ but not 200 µg $L^{-1}$.

**Table 2.** Summer (June–September) monthly median Beasley Lake, Mississippi, water quality variables used in determining Carlson-type trophic state index (TSI) scores [33–35] from 1998–2016. TSI scores include: Secchi visibility, TSI(SD); total phosphorus, TSI(TP); total nitrogen, TSI(TN); and chlorophyll *a*, TSI(Chl).

| Variable | June | July | August | September |
|---|---|---|---|---|
| | Median (Range) | Median (Range) | Median (Range) | Median (Range) |
| SD (m) | 0.42 (0.04–0.93) | 0.47 (0.05–1.07) | 0.45 (0.13–0.77) | 0.36 (0.06–0.90) |
| TP (mg $L^{-1}$) | 0.49 (0.06–3.65) | 0.37 (0.06–1.72) | 0.33 (0.05–0.85) | 0.38 (0.07–0.95) |
| TN (mg $L^{-1}$) | 1.46 (0.67–2.18) | 1.49 (0.82–7.02) | 1.28 (0.17–2.99) | 1.47 (0.79–3.32) |
| Chl (µg $L^{-1}$) | 33 (0–483) | 35 (0–147) | 31 (0–204) | 37 (0–257) |
| TSI(SD) | 72 (61–106) | 71 (59–103) | 72 (64–89) | 75 (62–101) |
| TSI(TP) | 93 (64–122) | 89 (63–112) | 88 (61–101) | 90 (66–103) |
| TSI(TN) | 60 (49–66) | 60 (52–83) | 58 (29–70) | 60 (51–72) |
| TSI(Chl) | 65 (0–91) | 65 (0–80) | 64 (0–83) | 66 (0–85) |

### 3.2. Temporal Changes in Trophic State

From 1998–2016, several Beasley Lake summer TSI indices exhibited significant temporal changes as shown in Table 3. The TSI(SD) and TSI(TP) significantly decreased over the 19-year period for each summer month with moderate [42] negative Mann–Kendall τ statistics ranging from −0.383 to −0.450 for TSI(SD) and −0.457 to −0.602 for TSI(TP). In contrast, TSI(Chl) trends were significantly positive for each month indicating increases in algal biomass during the 19-year study period. However, Mann–Kendall τ statistics for the months of June and July were weak to moderate ([42]; Table 3).

**Table 3.** Mann–Kendall Tau Statistic, τ, for Beasley Lake, Mississippi, summer (June–September) Carlson-type trophic state index (TSI) scores [33–35] versus time from 1998–2016. TSI scores include: Secchi visibility, TSI(SD); total phosphorus, TSI(TP); total nitrogen, TSI(TN); and chlorophyll *a*, TSI(Chl).

| Index | n | τ | n | τ | n | τ | n | τ |
|---|---|---|---|---|---|---|---|---|
| | | June | | July | | August | | September |
| TSI(SD) | 46 | **−0.391** [1] | 43 | **−0.450** | 40 | **−0.422** | 44 | **−0.383** |
| TSI(TP) | 46 | **−0.577** | 43 | **−0.602** | 41 | **−0.457** | 43 | **−0.548** |
| TSI(TN) | 34 | −0.014 | 31 | 0.166 | 31 | 0.163 | 32 | 0.133 |
| TSI(Chl) | 46 | **0.255** | 43 | **0.340** | 41 | **0.482** | 43 | **0.466** |
| TSI 1977 | 46 | **−0.364** | 43 | **−0.384** | 40 | **−0.313** | 43 | **−0.483** |
| TSI 1992 | 34 | −0.125 | 31 | −0.002 | 31 | −0.176 | 32 | −0.165 |
| TSI(Chl)–TSI(SD) | 46 | **0.563** | 43 | **0.543** | 40 | **0.512** | 43 | **0.516** |
| TSI(Chl)–TSI(TP) | 46 | **0.501** | 43 | **0.667** | 41 | **0.588** | 43 | **0.599** |
| TSI(Chl)–TSI(TN) | 34 | **0.316** | 31 | **0.295** | 31 | 0.231 | 32 | **0.319** |

[1] Bold τ values = slopes are statistically significantly different from zero ($p < 0.05$).

The TSI 1977 scores (comprised of TSI(SD), TSI(TP), and TSI(Chl)) significantly decreased for each summer month during the study period with scores between 75 and >90 from 1998 to 2001 and decreasing to <75 by 2003 (Figure 3a). This was represented by

negative Mann–Kendall τ statistics ranging from −0.313 in August to −0.483 in September (Table 3). Although statistically significant, Mann–Kendall τ statistics were weak to moderate [42], a result of most decreases in TSI 1977 occurring between 2001 and 2007. By 2008, TSI 1977 scores were similar across years and varied between low hypertrophic and high eutrophic states (Figure 3a). In comparison, the TSI 1992 scores (which included TSI(TN)) showed no significant temporal changes for any summer month during the study period (Table 3). No Mann–Kendall τ statistics were greater than 0.166. Most TSI 1992 scores ranged from 65 to 75 during years TN was measured indicating that with TN, trophic state was stable between low hypertrophic and high eutrophic states (Figure 3b).

Plotted median summer TSI deviation scores for all indices (Figure 4a,d and Figure 5a–d) indicated statistically significant temporal increases during summer except August TSI(Chl)–TSI(TN) (Table 3). Significant moderate increases in summer TSI(Chl)–TSI(SD) from 1998 to 2016 showed median scores moving from a low of −41 in June 1998 to the highest score of +11 in June 2016 with comparable increases in scores for every summer month (Figures 4 and 5).

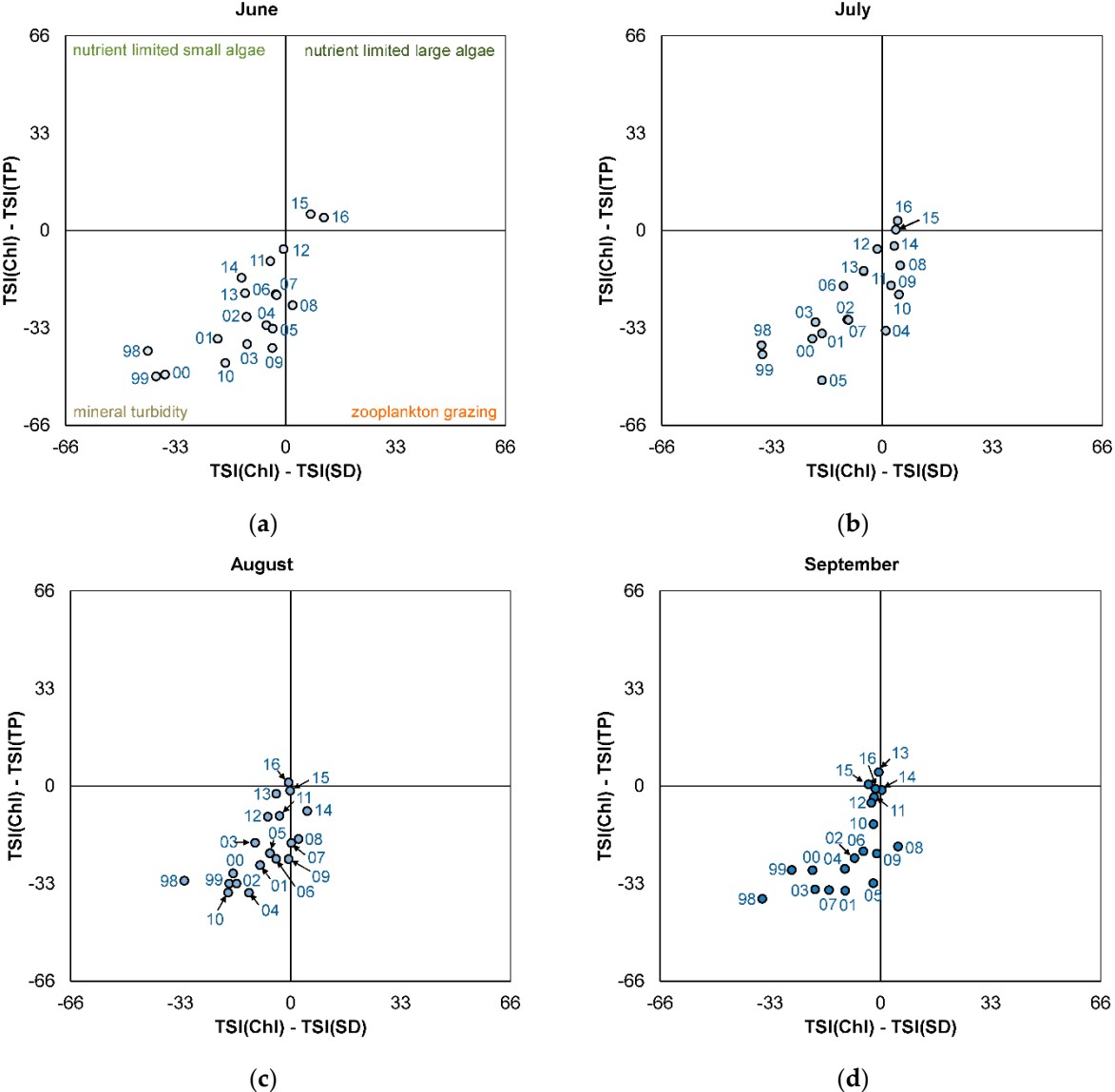

**Figure 4.** Beasley Lake, Mississippi, annual summer median Carlson-type 1977 [33] trophic state index (TSI) score deviation plots for: (**a**) June, (**b**) July, (**c**) August, and (**d**) September from 1998–2016 (98–16).

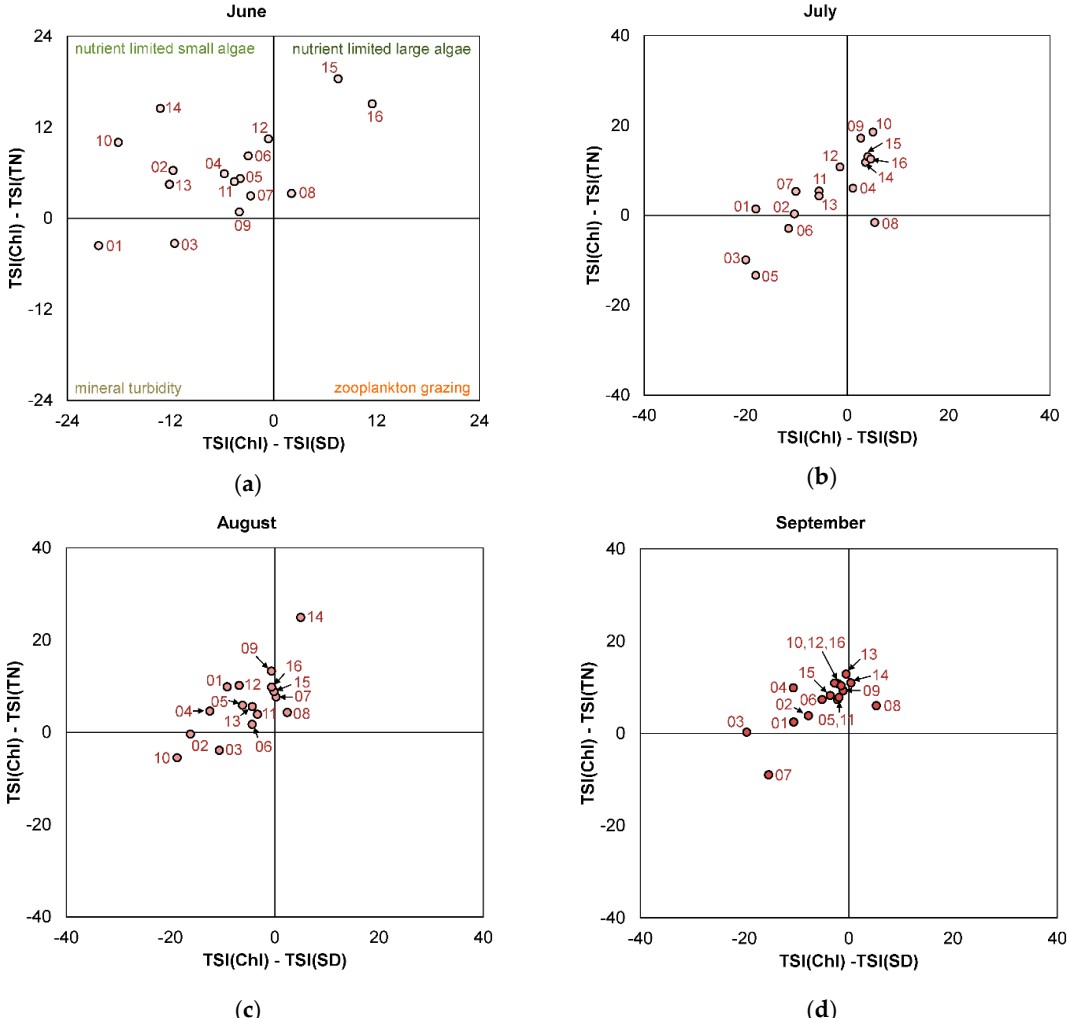

**Figure 5.** Beasley Lake, Mississippi, annual summer median Carlson-type 1992 [34,35] trophic state index (TSI) score deviation plots for: (**a**) June, (**b**) July, (**c**) August, and (**d**) September from 2001–2016 (01–16).

Mann–Kendall τ statistics were all greater than 0.5 (Table 3) and indicated a shift in the lake from light attenuation resulting from abiotic particles (e.g., suspended sediment) to light attenuation due to phytoplankton biomass [43]. Similarly, summer TSI(Chl)–SI(TP) scores moved from −49 in June 1999 to +5 in June 2015 and exhibited significant moderate increases in every month (Figure 4a–d) with Mann–Kendall τ statistics ranging from 0.501 in June to 0.667 in July (Table 3). Increases in TSI(Chl)–TSI(TP) scores indicate the lake could be moving towards phosphorus limitation [37]. In comparison, summer TSI(Chl)–TSI(TN) scores exhibited only modest increases with scores ranging from −13 in July 2005 to +25 in August 2014 (Figure 5a–d). Mann–Kendall τ statistics were significant for June, July, and September but not August and τ statistics were only weakly correlated with time (Table 3) [42]. Despite the relatively weaker associations of TSI(Chl)–TSI(TN) scores and time, increases in these scores indicate potential increase in lake nitrogen limitation [44].

### 3.3. Management Practices and Trophic State

Classification and regression tree models of trophic state and BMPs were generated for most summer months for every trophic state index but TSI(TN). CART models for summer TSI(SD) exhibited PRE values (comparable with multiple $R^2$ in regression models) from 0.676 in September to 0.858 in July with most change points resulting from implementation of 8.2–8.8 ha VBS in 2003–2004 (Table 4) and indicate significant decreases in summer non-algal light attenuation.

**Table 4.** Classification and regression tree (CART) analysis for summer (June–September) Carlson-type trophic state index trophic state index (TSI) scores [33–35] and best management practices (BMP) in Beasley Lake, Mississippi, from 1998–2016. TSI scores include: Secchi visibility, TSI(SD); total phosphorus, TSI(TP); total nitrogen, TSI(TN); and chlorophyll *a*, TSI(Chl). The CART model provides change points (point at which the independent variable separates the dependent variable into groups), proportional reduction in error (PRE) values (the model goodness-of-fit approximate to the coefficient of determination), and improvement values (the model individual independent variable goodness-of-fit).

| TSI | Node | Mean ± SD | Change Point | PRE | Improvement |
|---|---|---|---|---|---|
| TSI(SD) June | 1 | 78.5 ± 13.7 | 2004 VBS 8.2 ha | 0.822 | 0.822 |
| TSI(SD) July | 1 | 74.8 ± 10.7 | 2004 VBS 8.2 ha | 0.690 | 0.690 |
| | 2 | 90.1 ± 9.9 | 1999 CT 78.4 ha | 0.858 | 0.168 |
| TSI(SD) August | 1 | 73.1 ± 6.4 | 2004 VBS 8.2 ha | 0.641 | 0.641 |
| | 3 | 70.4 ± 3.7 | 2003 VBS 8.8 ha | 0.697 | 0.056 |
| TSI(SD) September | 1 | 76.0 ± 9.2 | 2004 VBS 8.2 ha | 0.676 | 0.676 |
| TSI(TP) June | 1 | 93.6 ± 11.7 | 2002 VBS 8.2 ha | 0.412 | 0.412 |
| | 3 | 83.0 ± 9.4 | 2009 CT 78.4 ha | 0.549 | 0.137 |
| TSI(TP) July | 1 | 88.2 ± 10.6 | 2006 QB 9 ha | 0.412 | 0.412 |
| | 2 | 95.3 ± 7.5 | 2000 CT 52.1 ha | 0.479 | 0.067 |
| | 3 | 81.8 ± 8.7 | 2010 SP 1 ha | 0.543 | 0.065 |
| TSI(TP) August | 1 | 85.5 ± 8.8 | 2010 SP 1 ha | 0.485 | 0.485 |
| | 2 | 89.7 ± 5.2 | 2000 CT 52.1 ha | 0.584 | 0.099 |
| | 3 | 76.3 ± 8.5 | 2016 CT 135.4 ha | 0.676 | 0.092 |
| | 4 | 78.6 ± 7.3 | 2004 CT 119 ha | 0.740 | 0.064 |
| TSI(TP) September | 1 | 87.1 ± 9.6 | 2010 SP 1 ha | 0.681 | 0.681 |
| TSI (Chl) June | 1 | 62.8 ± 12.5 | 2006 QB 9 ha | 0.064 | 0.064 |
| TSI (Chl) July | 1 | 63.4 ± 12.1 | 2006 QB 9 ha | 0.214 | 0.214 |
| | 2 | 57.6 ± 11.8 | 2003 CRP 87.3 ha | 0.273 | 0.058 |
| TSI (Chl) August | 1 | 61.7 ± 12.9 | 2006 QB 9 ha | 0.133 | 0.133 |
| | 2 | 57.0 ± 15.1 | 1998 CT 113 ha | 0.194 | 0.061 |
| | 5 | 53.8 ± 17.2 | 2001 CT 254.7 ha | 0.245 | 0.050 |
| TSI (Chl) September | 1 | 64.4 ± 10.9 | 2010 SP 1 ha | 0.194 | 0.194 |
| | 2 | 61.3 ± 11.5 | 2012 CT 25.7 ha | 0.412 | 0.218 |
| TSI 1977 June | 1 | 78 ± 8 | 2004 VBS 8.2 ha | 0.575 | 0.575 |
| TSI 1977 July | 1 | 75 ± 7 | 2004 VBS 8.2 ha | 0.359 | 0.359 |
| | 2 | 83 ± 7 | 2009 CT 78.4 ha | 0.463 | 0.103 |
| TSI 1977 August | 1 | 74 ± 5 | 2003 CRP 87.3 ha | 0.141 | 0.141 |
| | 2 | 76 ± 5 | 1998 CT 113 ha | 0.216 | 0.075 |
| TSI 1977 September | 1 | 76 ± 6 | 2003 CRP 87.3 ha | 0.367 | 0.367 |
| | 2 | 80 ± 4 | 2001 VBS 9.1 ha | 0.462 | 0.094 |
| | 3 | 73 ± 3 | 2012 CT 25.7 ha | 0.557 | 0.096 |
| | 6 | 68 ± 8 | 2010 SP 1 ha | 0.639 | 0.082 |
| TSI(Chl)–TSI(SD) June | 1 | −15.7 ± 20.4 | 2004 VBS 8.2 ha | 0.544 | 0.544 |
| | 2 | −40.3 ± 21.4 | 1998 CT 113 ha | 0.635 | 0.091 |
| TSI(Chl)–TSI(SD) July | 1 | −10.9 ± 16.4 | 2006 QB 9 ha | 0.407 | 0.407 |
| | 2 | −22.1 ± 15.8 | 2010 CT 119 ha | 0.520 | 0.112 |
| | 4 | −28.7 ± 12.6 | 2009 CT 78.4 ha | 0.580 | 0.060 |
| TSI(Chl)–TSI(SD) August | 1 | −11.4 ± 16.0 | 2006 QB 9 ha | 0.287 | 0.287 |
| | 2 | −20.2 ± 17.2 | 2002 VBS 8.2 ha | 0.360 | 0.073 |
| | 4 | −27.1 ± 18.4 | 1998 CT 113 ha | 0.457 | 0.097 |
| TSI(Chl)–TSI(SD) September | 1 | −11.9 ± 14.2 | 2004 VBS 8.2 ha | 0.379 | 0.379 |
| | 3 | −6.8 ± 12.3 | 2011 CTR 35 ha | 0.431 | 0.052 |
| | 4 | −12.1 ± 17.7 | 2010 SP 1 ha | 0.566 | 0.135 |

**Table 4.** *Cont.*

| TSI | Node | Mean ± SD | Change Point | PRE | Improvement |
|---|---|---|---|---|---|
| TSI(Chl)–TSI(TP) June | 1 | −30.8 ± 19.1 | 2002 VBS 8.2 ha | 0.280 | 0.280 |
| | 3 | −24.7 ± 14.3 | 1999 CT 78.4 ha | 0.358 | 0.078 |
| | 4 | −17.2 ± 13.3 | 2006 QB 9 ha | 0.422 | 0.063 |
| TSI(Chl)–TSI(TP) July | 1 | −24.8 ± 16.8 | 2006 QB 9 ha | 0.547 | 0.547 |
| | 3 | −13.1 ± 11.4 | 2010 SP 1 ha | 0.613 | 0.067 |
| TSI(Chl)–TSI(TP) August | 1 | −24.1 ± 16.8 | 2006 QB 9 ha | 0.284 | 0.284 |
| | 3 | −15.2 ± 13.5 | 2009 CT 78.4 ha | 0.351 | 0.067 |
| | 5 | −21.2 ± 15.5 | 2016 CT 135.4 ha | 0.410 | 0.059 |
| | 7 | −13.9 ± 12.0 | 2010 SP 1 ha | 0.471 | 0.061 |
| TSI(Chl)–TSI(TP) September | 1 | −22.7 ± 16.4 | 2010 SP 1 ha | 0.602 | 0.602 |
| TSI(Chl)–TSI(TN) June | 1 | 5.6 ± 7.9 | 2010 SP 1 ha | 0.103 | 0.103 |
| TSI(Chl)–TSI(TN) July | 1 | 4.3 ± 13.2 | 2006 QB 9 ha | 0.206 | 0.206 |
| | 2 | −5.2 ± 16.6 | 2002 VBS 8.8 ha | 0.324 | 0.118 |
| | 3 | 8.1 ± 9.4 | 2006 CT 161.9 ha | 0.384 | 0.061 |
| TSI(Chl)–TSI(TN) September | 1 | 5.2 ± 11.2 | 2012 CT 25.7 ha | 0.147 | 0.147 |
| | 2 | −6.1 ± 25.2 | 2010 SP 1 ha | 0.358 | 0.212 |
| | 3 | 6.8 ± 6.2 | 2006 QB 9 ha | 0.429 | 0.071 |

Summer TSI(TP) CART models exhibited PRE values from 0.543 in July to 0.740 in August with associated change points for varying BMPs: in June, primarily 8.2 ha VBS in 2002; July, primarily 9 ha QB in 2006; and August–September, primarily 1 ha SP in 2010 (Table 4). These varying BMPs decreased summer lake TSI(TP). Summer TSI(Chl) CART models were less robust with PRE values from 0.064 in June to 0.412 in September. The poor model fit for June precludes the use of this model for this month. BMP change points associated with TSI(Chl) for the remainder of the summer months varied: in July, 87.3 ha CRP and 9 ha QB in 2003 and 2006; August, 9 ha QB in 2003 and 113–254.7 ha CT in 1998, 2001; and September, 1 ha SP and 25.7 ha CT in 2010 and 2012. Implementation of these BMPs weakly to moderately increased TSI(Chl) for later summer months (July–September) during the study period. Summer TSI 1977 CART models produced PRE values that ranged widely from 0.216 in August to 0.639 in September with change points for primarily two BMPs: in June and July, implementation of 8.2 ha VBS in 2004; and in August and September, implementation of 87.3 ha CRP in 2003 (Table 4).

The CART models of summer TSI deviation scores and BMPs were generated for most summer months with the exception, again, of TN-derived scores (Table 4). Summer TSI(Chl)–TSI(SD) deviation CART models produced PRE values ranging from 0.457 in August to 0.635 in June. Models provided change points for the following BMPs: in June, 113 ha CT in 1998 and 8.2 ha VBS in 2004; in July, 9 ha QB in 2006 and 78.4–119 ha CT in 2009, 2010; in August, 113 ha CT in 1998, 8.2 ha VBS in 2002 and 9 ha QB in 2006; and September, 8.2 ha VBS in 2004, 1 ha SP in 2010 and 35 ha CT in 2011 (Table 4) and indicate these BMPs have helped reduced summer non-algal light attenuation (Figures 4 and 5). Summer TSI(Chl)–TSI(TP) deviation CART models produced similar PRE values ranging from 0.422 in June to 0.613 in July. The TSI(Chl)–TSI(TP) models provided the following BMP change points: in June, 78.4 ha CT in 1999, 8.2 ha VBS in 2002 and 9 ha QB 2006; in July, 9 ha QB in 2006 and 1 ha SP in 2010; in August, 9 ha QB in 2006, 78.4–135.4 ha CT in 2009 and 2016, and 1 ha SP in 2010; and in September, 1 ha SP in 2010 (Table 4) indicative of BMPs reducing P-laden suspended sediment and P becoming more closely linked with algal biomass (Figure 4a–d). As with other TN-derived scores, CART models of TSI(Chl)–TSI(TN) were limited. Summer TSI(Chl)–TSI(TN) model PRE values ranged from 0.103 in June to 0.429 in September with no model generated for August. Again, the poor model fit for June precludes the use of this model for this month. Model BMP change points for July were 8.8 ha VBS in 2002 and 161.9 ha CT with 9 ha QB in 2006. Model change points for September were 9 ha QB in 2006, 1 ha SP in 2010 and 25.7 ha CT in 2012 (Table 4).

These models indicate Beasley Lake could exhibit some increasing N limitation rather than P limitation during summer months.

## 4. Discussion

### 4.1. Lake Trophic Changes and Stable State

Several previous studies have demonstrated water quality improvement within Beasley Lake resulting from implementation of BMPs [23,24,26,28,45]. However, none of these studies has addressed whether these improvements have helped to mitigate eutrophication in the lake. Within the framework of the shallow lake system, limnologically, summer lake eutrophication (exclusive of TN) was moderately attenuated. Most of the attenuated trophic state occurred between 1998 and 2007 resulting from decreases in TP and increases in water clarity. Thereafter, the system appears to have reached a new phytoplankton-dominated stable state, or hysteresis [46], where continuing decreases in TP were offset by increases in algal biomass (as Chl) and no apparent increase in water clarity (Figure 4a–d). The juxtaposition of decreasing TP, increasing Chl and relatively unchanged water clarity could be attributed to changes in the phytoplankton community resulting from reduced suspended sediment loads and concomitant light limitation coinciding with decreases in TP loads. Havens et al. [47] studied a shallow lake system impacted by non-point source nutrient influx and observed that increased suspended sediment loads decreased light availability that favored cyanobacteria populations. Water clarity (i.e., light availability) is known to be influenced by cyanobacteria growth. Lakes with elevated TP and low water clarity favor cyanobacteria which out-compete other algae and cyanobacteria attenuate light greater than other groups of algae (e.g., chlorophytes and green algae) [48–50]. Within Beasley Lake, as TP loads decreased below approximately 0.5 mg/L during 2000–2002 and water clarity (Secchi visibility) increased to >0.4 m in 2004, this could have provided non-cyanobacteria algal populations the opportunity to begin to out-compete cyanobacteria and decrease the cyanobacteria populations in the lake.

Within the current stable state, further reductions in nutrients such as TP are likely to produce only modest (if any) increases in water clarity or decreases in algal biomass. For a shallow lake such as Beasley, only until nutrient concentrations occur below a certain threshold or change point will further improvements likely be manifested [51,52]. This indicates that Beasley Lake was altered from a predominantly light limited system impacted from elevated suspended sediment loads and possibly cyanobacteria to a stable, moderately less eutrophic phytoplankton-dominated system during the summer (Figure 4a–d). Although still classified as a eutrophic lake, Beasley Lake water clarity, nutrients and Chl measured within the last 3–4 years of the study (2013–2014) were increasingly similar to those observed in designated least-impaired oxbow lakes occurring in the lower Mississippi Alluvial Plain [11]. As a phytoplankton-dominated hysteretic system, Beasley Lake may need to significantly further reduce TP levels below some critical threshold [52,53] in order to nutrient-starve the system, decrease the intensity of algal blooms and summer algal biomass, and improve water clarity [6]. However, it is currently unclear what the nutrient threshold might be. Lower ranging TP concentrations for the last three years of the current study (2014–2016) ranged from 0.07 to 0.15 mg/L suggesting TP concentrations would need to be below 0.07 mg/L before TP becomes a more limiting nutrient.

### 4.2. Prospects for Further Improvements of Trophic State

The current study utilized multiple integrated BMPs throughout the watershed that, in theory and practice, reduce agricultural non-point source runoff TN and TP loads to Beasley Lake [28,30] which led to a modest decrease in trophic state. However, within-lake reductions in nutrients were apparent with mostly TP and dissolved inorganic N [23]. Previous research on lake restoration and improvement in trophic state through reductions in agricultural non-point sourced nutrients, wastewater sourced nutrients, biomanipulation via aquatic plants, or reducing fish populations has produced mixed results [6,15,16,54,55]. Most programs have focused on nutrient reductions with

some limited short-term (<10 years) success for TP control and mitigation [54] while less is understood about TN control [54,56]. However, these studies suggest controlling both TP and TN simultaneously have the best potential for success. Additionally, issues of lag-time between implementation of BMPs and complete responses in water quality variables [18] such as nutrients and water clarity with corresponding changes in algal biomass can make comprehensive assessments of trophic status changes more challenging.

Based on previous shallow lake restoration research, the potential for Beasley Lake to exhibit further improvements in trophic state appear to be limited. Shifting trophic state closer to a mesotrophic condition would require the following changes. Primarily the lake would need continued decreases in TP concentrations coupled with potential decreases in TN concentrations below some threshold that would nutrient starve the algal community, reduce algal biomass and further increase water clarity. To do this, determination of nutrient limitation and thresholds of the current algal community are necessary. Currently, despite decreases in Beasley Lake TP, with TN stable, concentrations of both nutrients indicate that phytoplankton within the system appear to not be immediately nutrient limited. Potential (Liebig) N-limitation has been observed [57] in Beasley Lake samples. However, this is likely secondary N-limitation resulting from TP concentrations remaining at levels that would not be limiting (e.g., >0.1 mg/L). These conditions were observed in a lake similarly impacted by non-point source agricultural pollution with elevated TN and TP concentrations [44]. This would indicate a clear need to further reduce and control both TN and TP concentrations as suggested by [56].

*4.3. Future Research Focus*

There remains a significant amount of information needed to assess if current BMPs in the watershed will elicit any further improvement in lake trophic status. Much less is known regarding the nutrient cycling and internal loading of both N and P in Beasley Lake. For example, the role of denitrification in the system to remove the pool of N and further decrease eutrophication needs to be determined. Limited available research indicates that denitrification potential exists within portions of the watershed draining into the lake, such as the extensive riparian habitat located to the east adjacent to the lake (Figure 1) and to a lesser extent within the upland cultivated soil of the watershed [58,59]. However, currently it is unknown how much, if any, denitrification occurs within the lake itself. This information would be critical to understanding the apparent stability of TN in the lake. For P, the role of internal loading in lake bed sediments and the rate at which P is released back into the lake and cycled out or flushed out of the lake needs to be understood [60]. The question of how long it would take to deplete the P pool in lake-bed sediments needs to be addressed to determine sustainable restoration [61–63]. Even if all external sources of N and P were captured and prevented from entering the lake, would this be sufficient to nutrient starve the lake system and cross a nutrient threshold to further decrease algal biomass and increase water clarity? Finally, what impact(s) to the lake system would result from global climate change? If precipitation (i.e., rainfall) events become increasingly more intense, even if total annual rainfall remains unchanged, would the system still be vulnerable to brief increases in nutrient and sediment loads from runoff and/or flooding from the adjacent river (Big Sunflower River), potentially hindering further improvements in trophic state? Recently, Yasarer et al. [64] indicated that climate change models predict such scenarios are possible within the watershed. However, these models also indicate that existing BMPs would still be efficient at mitigating nutrient and sediment runoff even with increased rainfall intensity.

## 5. Conclusions

One of the main values of agricultural best management practices (BMPs) demonstrated in the current study is to improve water quality and ecosystem services by decreasing nutrient enrichment or eutrophication through reductions in nutrient loads to the lake and lake surface water nutrients. While eutrophication can be measured by assessing

algae, water clarity, and nutrient levels, outside of the current study, documented changes in trophic status directly related to BMPs are still limited. The present study measured long-term (19-year) changes in summer eutrophication in an agriculturally influenced lake with a variety of BMPs placed in the watershed. Decreased eutrophication was observed as increasing water clarity (from <0.2 to >0.5 m) and decreasing total phosphorus (from >0.50 to <0.30 mg L$^{-1}$), over the first 10 years in association with increased vegetative buffers, a sediment retention pond, and conservation tillage. However, algae increased with conservation tillage and vegetative buffers while total nitrogen remained unchanged during the study. The study showed BMPs can modestly improve summer eutrophication, but these improvements can be offset by increases in algae. Future assessments in trophic status will need to consider other system complexities such as nutrient internal loading and cycling as well as global climate change. These results are valuable to regulatory and management agencies and farming stakeholders by providing additional information to improve and sustain water quality and overall environmental quality using BMPs.

**Author Contributions:** Conceptualization, R.E.L.J. and S.S.K.; methodology, R.E.L.J.; validation, R.E.L.J. and L.M.W.Y.; formal analysis, R.E.L.J.; investigation, R.E.L.J., M.A.L., and S.S.K.; resources, R.E.L.J. and S.S.K.; data curation, S.S.K. and L.M.W.Y.; writing—original draft preparation, R.E.L.J. and L.M.W.Y.; writing—review and editing, M.A.L., R.L.B., and S.S.K.; visualization, R.E.L.J.; supervision, M.A.L. and R.L.B.; project administration, M.A.L. and R.L.B. All authors have read and agreed to the published version of the manuscript.

**Funding:** This research received no external funding.

**Data Availability Statement:** Data for this study are available in a publicly accessible repository that does not issue DOIs. Publicly available datasets from a USDA-ARS web-based application, Sustaining the Earth's Watersheds, Agricultural Research Data System (STEWARDS) were used in this study. The STEWARDS v3.0 website database can be found here: https://www.nrrig.mwa.ars.usda.gov/stewards/stewards.html (accessed on 22 December 2020).

**Acknowledgments:** We extend our appreciation to Lisa Brooks, James Hill, and Sam Testa for chemical analyses and technical assistance. We also thank Wade Steinriede, Calvin Vick, and Mark Griffith for sample collection and maintenance of land management information. Disclaimer: The use of trade, firm, or corporation names is solely for the information and convenience of the reader. Mention of names does not constitute an official endorsement or approval by the USDA or the Agricultural Research Service of any product or service to the exclusion of others that may be suitable. The USDA prohibits any discrimination in all its programs and activities on the basis of race, color, national origin, age, disability, sex, marital status, familial status, parental status, religion, sexual orientation, genetic information, political beliefs, reprisal, or because any part of an individual's income is derived from any public assistance program.

**Conflicts of Interest:** The authors declare no conflict of interest.

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
