# Peer review of "Long-Term Oxbow Lake Trophic State under Agricultural Best Management Practices"

_water, doi:10.3390/w13081123_

Round 1
Reviewer 1 Report
Dear Authors,
I am writing this to submit my comments on your research article with the following details.
Manuscript title: Long-Term Oxbow Lake Trophic State Under Agricultural Best Management practices
Manuscript Number: water-1171380
Journal Submitted: Water
Specific Comments:
Title:
The title is acceptable.
Abstract:
L 19-22. I guess you have repeated the same thing with a bit of change. Please look into this.
It would be best if you mentioned a bit of the BMP adopted for this study or considered designing it.
Please describe a bit of your study design in the beginning.
The results are described in a way that the abstract looks a generic tale. Please enrich with more specific findings.
Please improve your conclusions and opening.
Introduction:
L 34-37: Reference is missing.
The introduction is shallow. Please add more specific details on TSI and past applications of BMP.
Also, provide the rationale why you took this study.
Please improve your objectives.
Materials and Methods:
The methods are given with sufficient details that may help future researchers to follow.
Results:
L 190: indexes or indices?
The results are written well.
Discussion:
This is perfectly well-written.
Figures and Tables:
Figure 2. Why is there a huge discrepancy in the CTA during the study period?
Table 2. This is too long. Please rearrange it in a way that it takes less space. Also, you may add min. and max in the same row divided into two rows. You must consider deleting the 25th and 75th % columns. Try to arrange the adjacent months' columns.
Figure 4. Please try to make it a more colorful figure.
Figure 5. Same as for figure 4.
Conclusions
Missing.
References:
Acceptable.
Author Response
Special Issues Journal Water (ISSN 2073-4441)
Manuscript ID water-1171380
Title
Long-Term Oxbow Lake Trophic State Under Agricultural Best Management practices
Authors
Richard E. Lizotte * , Lindsey M.W. Yasarer , Ronald L. Bingner , Martin A. Locke , and Scott S. Knight
Please provide a point-by-point response to the reviewer’s comments and either enter it in the box below or upload it as a Word/PDF file. Please write down "Please see the attachment." in the box if you only upload an attachment. An example can be found here.
* Author's Notes to Reviewer
Reviewer 1
Author's Reply to the Review Report in Bold Font (Reviewer 1)
Open Review
(x) I would not like to sign my review report
( ) I would like to sign my review report
English language and style
( ) Extensive editing of English language and style required
(x) Moderate English changes required
( ) English language and style are fine/minor spell check required
( ) I don't feel qualified to judge about the English language and style
Yes
Can be improved
Must be improved
Not applicable
Does the introduction provide sufficient background and include all relevant references?
( )
( )
(x)
( )
Is the research design appropriate?
( )
(x)
( )
( )
Are the methods adequately described?
( )
(x)
( )
( )
Are the results clearly presented?
(x)
( )
( )
( )
Are the conclusions supported by the results?
( )
( )
(x)
( )
Comments and Suggestions for Authors
Dear Authors,
I am writing this to submit my comments on your research article with the following details.
Manuscript title: Long-Term Oxbow Lake Trophic State Under Agricultural Best Management practices
Manuscript Number: water-1171380
Journal Submitted: Water
Specific Comments:
Title:
The title is acceptable.
The authors thank the reviewer for reviewing the title.
Abstract:
L 19-22. I guess you have repeated the same thing with a bit of change. Please look into this.
It would be best if you mentioned a bit of the BMP adopted for this study or considered designing it.
Page 1, Lines 15-17: The authors have included a statement of the BMPs as follows: ‘…Structural BMPs included vegetative buffers, conservation tillage, conservation reserve, a constructed wetland, and a sediment retention pond…’
Please describe a bit of your study design in the beginning.
Page 1, lines 13-18: The authors have revised and rewritten two sentences to provide more details to the study design within the first three sentences of the abstract as follows: ‘…Long-term (1998-2016) lake summer trophic state index (TSI) trends of an agricultural watershed with agricultural best management practices (BMPs) were assessed. Structural BMPs included vegetative buffers, conservation tillage, conservation reserve, a constructed wetland, and a sediment retention pond. TSI included Secchi visibility (SD), chlorophyll a (Chl), total phosphorus (TP) and total nitrogen (TN)…’
The results are described in a way that the abstract looks a generic tale. Please enrich with more specific findings.
Page 1, lines 18-23: the authors have provided more details regarding the study results as follows: ‘…Summer TSI 1977 was >80 in 1998-1999 (hypertrophic) and decreased over the first 10 years to TSI 1977 ≈75 (eutrophic). TSI 1977 decrease and changing TSI deviations coincided with vegetative buffers, conservation tillage, and conservation reserve. TSI(SD) decrease (>90 to <70) coincided with vegetative buffers and TSI(TP) decrease (>90 to <75) coincided primarily with conservation tillage and the sediment retention pond. TSI(Chl) increase (<60 to >70) coincided with conservation tillage and vegetative buffer…’
Please improve your conclusions and opening.
Page 1, lines 12-13: The authors have attempted to improve the opening as follows: ‘…A key principle of agricultural best management practices (BMPs) is to improve water quality by reducing agricultural-sourced nutrients and associated eutrophication…’
Page 1, lines 23-27: The authors have attempted to improve the conclusions as follows: ‘…Results indicate watershed-wide BMPs can modestly decrease summer trophic state through increased water transparency and decreased TP, but these changes are off-set by increases in chlorophyll a to reach a new stable state within a decade. Future research should assess algal nutrient thresholds, internal nutrient loading, and climate change effects…’
Authors note to the reviewer and editor: the journal requires that the abstract not exceed 200 words. This required that the abstract be completely revised and rewritten to attempt to incorporate more information on the study design, more specific results, and improved introduction (opening) and conclusions. But this limits any greater details from being included in the abstract.
Introduction:
L 34-37: Reference is missing.
Page 1, Line 37: The authors have included the appropriate references for this statement as follows:
[1,4,6]
- Jeppesen, E.; Søndergaard, M.; Lauridsen, T.L.; Kronvang, B.; Beklioglu, M.; Lammens, E.; Jensen, H.S.; Köhler, J.; Ventalä, M.; Tarvainen, M.; Tátrai, I. Danish and other European experiences in managing shallow lakes. Lake Reserv. Manage. 2007, 23, 439-451. http://dx.doi.org/10.1080/07438140709354029
- Liu, W.; Zhang, Q.; Liu, G. Influences of watershed landscape composition and configuration on lake-water quality in the Yangtze River basin of China. Hydrol. Process. 2012, 26, 570-578. DOI: 10.1002/hyp.8157
- Cooke, G.D. History of eutrophic lake rehabilitation in North America with arguments for including social sciences in the paradigm. Lake Reserv. Manage. 2007, 23, 323-329. DOI: 10.1080/07438140709354021
Page 1, Line 41, Line 43: Additionally, the authors have renumbered the following references from [6] to [7] on line 41 and from [7] to [6] on line 43.
The introduction is shallow. Please add more specific details on TSI and past applications of BMP.
Also, provide the rationale why you took this study.
Page 2, Lines 63-71: The authors have provided more specific details on the use of TSI studies in lake watersheds as follows:
‘…A useful tool for lake eutrophication assessment that incorporates the relationship between nutrients and algal blooms has been the Carlson-type trophic state index (TSI) [7] used in several lake studies over the last two decades [3-4,11-17]. The index is valuable because calculation requires relatively few commonly measured lake water quality variables: Secchi depth visibility, nutrients (nitrogen, phosphorus), and chlorophyll a [3-4,7]. TSI has been utilized for lake management of water use (e.g., drinking water, fish and wildlife use) [11-12], assist in lake ecoregional nutrient criteria development [13-14], assess the impacts of land-use, such as agriculture, on lake trophic state [3-4,15], and assess lake restoration success or recovery from storm events [16-17]…’
Page 18, Lines 490-506: The authors have also revised the References section to include new references and renumbered current references to reflect the changes made to the Introduction section as follows:
- Justus, B. Water quality of least-impaired lakes in eastern and southern Arkansas. Environ. Monit. Assess. 2010, 168, 363-383. DOI 10.1007/s10661-009-1120-5
- Knowlton, M.F.; Jones, J.R. Temporal variation and assessment of trophic state indicators in Missouri reservoirs: Implication for lake monitoring and management. Lake Reserv. Manage. 2006, 22, 261-271.
- Dodds, W.K. Determining ecoregional reference conditions for nutrients, Secchi depth and chlorophyll a in Kansas lakes and reservoirs. Lake Reserv. Manage. 2006, 22, 151-159.
- Jones, J.R.; Obrecht, D.V.; Perkins, B.D.; Knowlton, M.F.; Thorpe, A.P.; Watanabe, S.; Bacon, R.R. Nutrients, seston, and transparency of Missouri reservoirs and oxbow lakes: An analysis of regional limnology. Lake Reserv. Manage. 2008, 24, 155-180.
- Thomatou, A.A.; Triantafyllidou, M.; Chalkia, E.; Kehayias, G.; Konstantinou, I.; Zacharias, I. Land use changes do not rapidly change the trophic state of a deep lake. Amvrakia Lake, Greece. Journal of Environmental Protection 2013, 4, 426-434. doi:10.4236/jep.2013.45051
- Poor, N.D. Effect of lake management efforts on the trophic state of a subtropical shallow lake in Lakeland, Florida, USA. Water Air Soil Pollut. 2010, 207, 333-347. DOI 10.1007/s11270-009-0140-7
- Havens, K.; Ji, G. Inferences about seston composition and phytoplankton limiting factors during recovery of a large shallow lake from hurricane impacts. Inland Waters, 2017, 7, 236-247. https://doi.org/10.1080/20442041.2017.1320902
Page 2, Lines 72-78: The authors have provided more specific details on past applications of BMPs in watersheds, specifically in relation to mitigating eutrophication as follows:
‘…Agricultural best management practices (BMPs) have been implemented globally as a mechanism for managing soil and water resources while maintaining or improving agricultural production [18-20]. Several of these same practices have the potential to mitigate impacts of agricultural activity on receiving water bodies such as lakes [21-22]. Proper water resource management, such as water quality, can be aided by BMPs that can intercept, mitigate, and process agricultural contaminants such as nutrients [18,20]. However, there is limited information on the effectiveness of BMPs in controlling eutrophication in agricultural watershed lakes [21-24]…’
Page 18, Lines 514-532: The authors have also Revised the reference section to include new references and renumbered current references to reflect the changes made to the Introduction section as follows:
- Meals, D.W.; Dressing, S.A.; Davenport, T.E. Lag time in water quality response to best management practices: A review. J. Environ. Qual. 2010, 39, 85-96. doi:10.2134/jeq2009.0108
- Tuppad, P.; Santhi, C.; Srinivasan, R. Assessing BMP effectiveness: multiprocedure analysis of observed water quality data. Environ. Monit. Assess. 2010, 170, 315-329. DOI 10.1007/s10661-009-1235-8
- Zhang, X.; Liu, X.; Zhang, M.; Dahlgren, R.A.; Eitzel, M. A review of vegetated buffers and a meta-analysis of their mitigation efficacy in reducing nonpoint source pollution. J. Environ. Qual. 2010, 39, 76-84. doi:10.2134/jeq2008.0496
- Payne, F.E.; Bjork, T.M. The effectiveness of BMPs and sediment control structures and their relationship to in-lake water quality. Lake Reserv. Manage. 1984, 1, 82-86.
- Makarewicz, J.C.; Lewis, T.W.; Bosch, I.; Noll, M.R.; Herendeen, N.; Simon, R.D.; Zollweg, J.; Vodacek, A. The impact of agricultural best management practices on downstream systems: Soil loss and nutrient chemistry and flux to Conesus Lake, New York, USA. J. Great Lakes Res. 2009, 35, 23-36. doi:10.1016/j.jglr.2008.10.006
- Lizotte, R.E.; Yasarer, L.M.W.; Locke, M.A.; Bingner, R.L.; Knight, S.S. Lake nutrient responses to integrated conservation best management practices in an agricultural watershed. J. Environ. Qual. 2017, 46, 330-338. doi:10.2134/jeq2016.08.0324
- Locke, M.; Knight, S.; Smith, S.; Cullum, R.; Zablotowicz, R.; Yuan, Y.; Bingner, R.L. Environmental quality research in the Beasley Lake watershed, 1995 to 2007: succession from conventional to conservation practices. J. Soil Water Conserv. 2008, 63, 430-442. doi:10.2489/jswc.63.6.430
Please improve your objectives.
Page 3, Lines 87-93: The authors have revised the study objectives to better clarify the study goals as follows:
‘…Objectives of the current study are to assess the following: 1) what is the long-term (19-year) summer trophic state of an agriculturally influenced oxbow lake, Beasley Lake; 2) has summer lake trophic state changed during the long-term assessment period; 3) are there any trends in any changes to summer lake trophic state during the study period; 4) are these trends in altered summer lake trophic state associated with multiple BMPs implemented in the watershed; and 5) are these associations between altered summer lake trophic state and implemented watershed BMPs indicative of mitigated eutrophication?..’
Materials and Methods:
The methods are given with sufficient details that may help future researchers to follow.
The authors thank the reviewer for reviewing the Materials and Methods.
Results:
L 190: indexes or indices?
Although both indexes and indices are used in the English language, the more formal ‘indices’ is more appropriate for scientific contexts. For this reason, the authors have revised ‘indexes’ to ‘indices’.
Page 8, Line 239: ‘indexes’ changed to ‘indices’
Page 9, Line 263: ‘indexes’ changed to ‘indices’
The results are written well.
The authors thank the reviewer for reviewing the results.
Discussion:
This is perfectly well-written.
The authors thank the reviewer for reviewing the Discussion.
Figures and Tables:
Figure 2. Why is there a huge discrepancy in the CTA during the study period?
The authors note that conservation tillage practices are not permanent structural conservation practices but cultural ones that are decided by individual farmers from year to year in the watershed. As such, the areas of conservation tillage are variable. To clarify this distinction for the reader, this information is included in the body of the manuscript as follows:
Page 4, Lines 128-131: ‘…Conservation tillage practices are not permanent structural conservation practices but cultural ones that are decided by individual farmers from year to year in the watershed. As such, the areas of conservation tillage are variable…’
Table 2. This is too long. Please rearrange it in a way that it takes less space. Also, you may add min. and max in the same row divided into two rows. You must consider deleting the 25th and 75th % columns. Try to arrange the adjacent months' columns.
Page 7, Line 232: The authors have rearranged the table columns and rows in Table 2 and have significantly shortened the table so that the table now take less space. As suggested by the reviewer, the 25th and 75th % columns have been deleted. To allow for all measured data and ranges (minimum to maximum values) to be included in the same row, the sample size data also had to be deleted. The table has been rearranged as follows.
From:
Variable |
n |
Median |
25th-75th % |
Minimum |
Maximum |
|
June |
||||
Secchi Visibility (m) |
133 |
0.42 |
0.10-0.60 |
0.04 |
0.93 |
Total Phosphorus (mg L-1) |
133 |
0.49 |
0.30-0.93 |
0.06 |
3.65 |
Total Nitrogen (mg L-1) |
94 |
1.46 |
1.16-1.71 |
0.67 |
2.18 |
Chlorophyll a (μg L-1) |
133 |
33 |
17-48 |
0 |
483 |
TSI(SD) |
133 |
72 |
67-93 |
61 |
106 |
TSI(TP) |
133 |
93 |
86-103 |
64 |
122 |
TSI(TN) |
94 |
60 |
57-62 |
49 |
66 |
TSI(Chl) |
133 |
65 |
58-69 |
0 |
91 |
|
July |
||||
Secchi Visibility (m) |
132 |
0.47 |
0.32-0.55 |
0.05 |
1.07 |
Total Phosphorus (mg L-1) |
131 |
0.37 |
0.21-0.55 |
0.06 |
1.72 |
Total Nitrogen (mg L-1) |
92 |
1.49 |
1.18-1.88 |
0.82 |
7.02 |
Chlorophyll a (μg L-1) |
132 |
35 |
16-72 |
0 |
147 |
TSI(SD) |
132 |
71 |
69-76 |
59 |
103 |
TSI(TP) |
131 |
89 |
81-95 |
63 |
112 |
TSI(TN) |
92 |
60 |
57-64 |
52 |
83 |
TSI(Chl) |
132 |
65 |
58-73 |
0 |
80 |
|
August |
||||
Secchi Visibility (m) |
124 |
0.45 |
0.33-0.58 |
0.13 |
0.77 |
Total Phosphorus (mg L-1) |
127 |
0.33 |
0.20-0.44 |
0.05 |
0.85 |
Total Nitrogen (mg L-1) |
97 |
1.28 |
1.04-1.75 |
0.17 |
2.99 |
Chlorophyll a (μg L-1) |
127 |
31 |
20-44 |
0 |
204 |
TSI(SD) |
124 |
72 |
68-76 |
64 |
89 |
TSI(TP) |
127 |
88 |
81-92 |
61 |
101 |
TSI(TN) |
97 |
58 |
55-63 |
29 |
70 |
TSI(Chl) |
127 |
64 |
60-68 |
0 |
83 |
|
September |
||||
Secchi Visibility (m) |
132 |
0.36 |
0.22-0.52 |
0.06 |
0.90 |
Total Phosphorus (mg L-1) |
133 |
0.38 |
0.18-0.53 |
0.07 |
0.95 |
Total Nitrogen (mg L-1) |
96 |
1.47 |
1.21-1.79 |
0.79 |
3.32 |
Chlorophyll a (μg L-1) |
134 |
37 |
20-59 |
0 |
257 |
TSI(SD) |
132 |
75 |
70-82 |
62 |
101 |
TSI(TP) |
133 |
90 |
79-95 |
66 |
103 |
TSI(TN) |
96 |
60 |
57-63 |
51 |
72 |
TSI(Chl) |
134 |
66 |
60-71 |
0 |
85 |
To:
|
June |
July |
August |
September |
Variable |
Median (Range) |
Median (Range) |
Median (Range) |
Median (Range) |
SD (m) |
0.42 (0.04-0.93) |
0.47 (0.05-1.07) |
0.45 (0.13-0.77) |
0.36 (0.06-0.90) |
TP (mg L-1) |
0.49 (0.06-3.65) |
0.37 (0.06-1.72) |
0.33 (0.05-0.85) |
0.38 (0.07-0.95) |
TN (mg L-1) |
1.46 (0.67-2.18) |
1.49 (0.82-7.02) |
1.28 (0.17-2.99) |
1.47 (0.79-3.32) |
Chl (μg L-1) |
33 (0-483) |
35 (0-147) |
31 (0-204) |
37 (0-257) |
TSI(SD) |
72 (61-106) |
71 (59-103) |
72 (64-89) |
75 (62-101) |
TSI(TP) |
93 (64-122) |
89 (63-112) |
88 (61-101) |
90 (66-103) |
TSI(TN) |
60 (49-66) |
60 (52-83) |
58 (29-70) |
60 (51-72) |
TSI(Chl) |
65 (0-91) |
65 (0-80) |
64 (0-83) |
66 (0-85) |
Figure 4. Please try to make it a more colorful figure.
Page 10, Lines 270-271: The authors have attempted to make the figure more colorful by using different color fills for the circles and using color fonts for the years shown in the figure.
Figure 5. Same as for figure 4.
Pages 12, Lines 279-280: The authors have attempted to make the figure more colorful by using different color fills for the circles and using color fonts for the years shown in the figure.
Conclusions
Missing.
Pages 16-17, Lines 432-449: While the journal does not require conclusions, because the discussion is somewhat complex, the authors agree to include a brief conclusion section. The section is as follows.
- Conclusions
One of the main values of agricultural best management practices (BMPs) demonstrated in the current study is to improve water quality and ecosystem services by decreasing nutrient enrichment or eutrophication through reductions in nutrient loads to the lake and lake surface water nutrients. While eutrophication can be measured by assessing algae, water clarity and nutrient levels, outside of the current study, documented changes in trophic status directly related to BMPs are still limited. The present study measured long-term (19-year) changes in summer eutrophication in an agriculturally influenced lake with a variety of BMPs placed in the watershed. Decreased eutrophication was observed as increasing water clarity (from <0.2 to >0.5 m) and decreasing total phosphorus (from >0.50 to <0.30 mg L-1), over the first 10 years in association with increased vegetative buffers, a sediment retention pond and conservation tillage. However, algae increased with conservation tillage and vegetative buffers while total nitrogen remained unchanged during the study. The study showed BMPs can modestly improve summer eutrophication, but these improvements can be offset by increases in algae. Future assessments in trophic status will need to consider other system complexities such as nutrient internal loading and cycling as well as global climate change. These results are valuable to regulatory and management agencies and farming stakeholders by providing additional information to improve and sustain water quality and overall environmental quality using BMPs.
References:
Acceptable.
The authors thank the reviewer for reviewing the References.
Reviewer 2 Report
The article is very interesting - presents an analysis of the impact of watershed reclamation on the trophic state of a small oxbow lake over many years. I have no objections to the methods and statistical analyses used in the paper.
I am not surprised by the lack of clear ecosystem response (trophic state improvement) of a shallow water body after reducing allochthonous nitrogen and phosphorus loads. Many years of accumulation of agricultural nutrient loads resulted in the formation of fertile sediments. Lack of thermal stratification and mixing of waters causes resuspension of nitrogen and phosphorus from the sediments and trigger primary production. Such water bodies keep high trophic state long time after the allochthonous nitrogen and phosphorus loads have been reduced. It seems that with such a large internal supply, the only solution is to control not the amount of primary production but rather the type of producers - the transition from a state where algae dominates to a state where primary production run by macrophytes. Many literature data and my observations indicate that it can be done by top-down regulation (strengthening the population of predatory fish, or even temporarily removing fish from the ecosystem).
Information on crop fertilization is worth adding.
line 92 instead of: “acreage of watershed” give “watershed area”
Authors can find some information about nutrients resuspension in papers published by GoÅ‚dyn R., e.g “Internal phosphorus loading as the response to complete and then limited sustainable restoration of a shallow lake”
Author Response
Special Issues Journal Water (ISSN 2073-4441)
Manuscript ID water-1171380
Title
Long-Term Oxbow Lake Trophic State Under Agricultural Best Management practices
Authors
Richard E. Lizotte * , Lindsey M.W. Yasarer , Ronald L. Bingner , Martin A. Locke , and Scott S. Knight
Please provide a point-by-point response to the reviewer’s comments and either enter it in the box below or upload it as a Word/PDF file. Please write down "Please see the attachment." in the box if you only upload an attachment. An example can be found here.
* Author's Notes to Reviewer
Reviewer 2
Author's Reply to the Review Report in Bold Font (Reviewer 2)
Review Report Form
Open Review
(x) I would not like to sign my review report
( ) I would like to sign my review report
English language and style
( ) Extensive editing of English language and style required
( ) Moderate English changes required
(x) English language and style are fine/minor spell check required
( ) I don't feel qualified to judge about the English language and style
Yes
Can be improved
Must be improved
Not applicable
Does the introduction provide sufficient background and include all relevant references?
(x)
( )
( )
( )
Is the research design appropriate?
(x)
( )
( )
( )
Are the methods adequately described?
(x)
( )
( )
( )
Are the results clearly presented?
(x)
( )
( )
( )
Are the conclusions supported by the results?
(x)
( )
( )
( )
Comments and Suggestions for Authors
The article is very interesting - presents an analysis of the impact of watershed reclamation on the trophic state of a small oxbow lake over many years. I have no objections to the methods and statistical analyses used in the paper.
I am not surprised by the lack of clear ecosystem response (trophic state improvement) of a shallow water body after reducing allochthonous nitrogen and phosphorus loads. Many years of accumulation of agricultural nutrient loads resulted in the formation of fertile sediments. Lack of thermal stratification and mixing of waters causes resuspension of nitrogen and phosphorus from the sediments and trigger primary production. Such water bodies keep high trophic state long time after the allochthonous nitrogen and phosphorus loads have been reduced. It seems that with such a large internal supply, the only solution is to control not the amount of primary production but rather the type of producers - the transition from a state where algae dominates to a state where primary production run by macrophytes. Many literature data and my observations indicate that it can be done by top-down regulation (strengthening the population of predatory fish, or even temporarily removing fish from the ecosystem).
Information on crop fertilization is worth adding.
Page 4, Lines 131-136: The authors have included crop fertilization information in the text of the manuscript and referenced a revised figure 2 to include a new figure 2b that includes fertilizer information as follows:
‘…Similarly, fertilizer application decisions were made by farmers and typically included applications added in spring months (March-June) prior to planting a crop, and was usually knifed in. Nitrogen fertilizer, primarily as urea-ammonium-nitrate, was applied every year except 2008-2010. Phosphorus fertilizer was only occasionally applied with the largest application occurring in 2005. Potassium and sulfur were applied frequently from 1998-2005, but only occasionally thereafter (Figure 2b)…’
Page 4, Lines 137-140: The authors have included crop fertilization rate information for the watershed for the study years as a new figure ‘Figure 2b‘ adjacent to revised figure ‘Figure 2a’ (conservation tillage).
line 92 instead of: “acreage of watershed” give “watershed area”
Page 4, Line 128: Correction made by the authors from ‘acreage of the watershed’ to ‘watershed area’.
Authors can find some information about nutrients resuspension in papers published by GoÅ‚dyn R., e.g “Internal phosphorus loading as the response to complete and then limited sustainable restoration of a shallow lake”
Page 16, Lines 421-422: The authors thank the reviewer for bringing to their attention the work by Ryszard GoÅ‚dyn and associates. The authors have included references to their work on phosphorus internal loading and sustainable lake restoration as follows: ‘…The question of how long it would take to deplete the P pool in lake-bed sediments needs to be addressed to determine sustainable restoration [61-63]…’
Page 20, Lines 638-646: The authors have included the new Ryszard GoÅ‚dyn and associates’ references [54-56] in the revised References section as follows:
- GoÅ‚dyn, R.; PodsiadÅ‚owski, S.; Dondajewska, R.; Kozak, A. The sustainable restoration of lakes—towards the challenges of the Water Framework Directive. Ecohydrol. Hydrobiol. 2014, 14, 68-74. http://dx.doi.org/10.1016/j.ecohyd.2013.12.001
- Dondajewska, R.; Kozak, A.; Kowalczewska-Madura, K.; Budzyńska, A.; Gołdyn, R.; Podsiadłowski, S.; Tomkowiak, A. The response of a shallow hypertrophic lake to innovative restoration measures - Uzarzewskie Lake case study. Ecol. Eng. 2018, 121, 72-82. https://doi.org/10.1016/j.ecoleng.2017.07.010
- Kowalczewska-Madura, K.; Dondajewska, R.; Gołdyn, R.; Rosińska, J.; Podsiadłowski, S. Internal phosphorus loading as the response to complete and then limited sustainable restoration of a shallow lake. Ann. Limnol. - Int. J. Lim. 2019, 55, 4. https://doi.org/10.1051/limn/2019003
Special Issues Journal Water (ISSN 2073-4441)
Manuscript ID water-1171380
Title
Long-Term Oxbow Lake Trophic State Under Agricultural Best Management practices
Authors
Richard E. Lizotte * , Lindsey M.W. Yasarer , Ronald L. Bingner , Martin A. Locke , and Scott S. Knight
Please provide a point-by-point response to the reviewer’s comments and either enter it in the box below or upload it as a Word/PDF file. Please write down "Please see the attachment." in the box if you only upload an attachment. An example can be found here.
* Author's Notes to Reviewer
Reviewer 2
Author's Reply to the Review Report in Bold Font (Reviewer 2)
Review Report Form
Open Review
(x) I would not like to sign my review report
( ) I would like to sign my review report
English language and style
( ) Extensive editing of English language and style required
( ) Moderate English changes required
(x) English language and style are fine/minor spell check required
( ) I don't feel qualified to judge about the English language and style
Yes
Can be improved
Must be improved
Not applicable
Does the introduction provide sufficient background and include all relevant references?
(x)
( )
( )
( )
Is the research design appropriate?
(x)
( )
( )
( )
Are the methods adequately described?
(x)
( )
( )
( )
Are the results clearly presented?
(x)
( )
( )
( )
Are the conclusions supported by the results?
(x)
( )
( )
( )
Comments and Suggestions for Authors
The article is very interesting - presents an analysis of the impact of watershed reclamation on the trophic state of a small oxbow lake over many years. I have no objections to the methods and statistical analyses used in the paper.
I am not surprised by the lack of clear ecosystem response (trophic state improvement) of a shallow water body after reducing allochthonous nitrogen and phosphorus loads. Many years of accumulation of agricultural nutrient loads resulted in the formation of fertile sediments. Lack of thermal stratification and mixing of waters causes resuspension of nitrogen and phosphorus from the sediments and trigger primary production. Such water bodies keep high trophic state long time after the allochthonous nitrogen and phosphorus loads have been reduced. It seems that with such a large internal supply, the only solution is to control not the amount of primary production but rather the type of producers - the transition from a state where algae dominates to a state where primary production run by macrophytes. Many literature data and my observations indicate that it can be done by top-down regulation (strengthening the population of predatory fish, or even temporarily removing fish from the ecosystem).
Information on crop fertilization is worth adding.
Page 4, Lines 131-136: The authors have included crop fertilization information in the text of the manuscript and referenced a revised figure 2 to include a new figure 2b that includes fertilizer information as follows:
‘…Similarly, fertilizer application decisions were made by farmers and typically included applications added in spring months (March-June) prior to planting a crop, and was usually knifed in. Nitrogen fertilizer, primarily as urea-ammonium-nitrate, was applied every year except 2008-2010. Phosphorus fertilizer was only occasionally applied with the largest application occurring in 2005. Potassium and sulfur were applied frequently from 1998-2005, but only occasionally thereafter (Figure 2b)…’
Page 4, Lines 137-140: The authors have included crop fertilization rate information for the watershed for the study years as a new figure ‘Figure 2b‘ adjacent to revised figure ‘Figure 2a’ (conservation tillage).
line 92 instead of: “acreage of watershed” give “watershed area”
Page 4, Line 128: Correction made by the authors from ‘acreage of the watershed’ to ‘watershed area’.
Authors can find some information about nutrients resuspension in papers published by GoÅ‚dyn R., e.g “Internal phosphorus loading as the response to complete and then limited sustainable restoration of a shallow lake”
Page 16, Lines 421-422: The authors thank the reviewer for bringing to their attention the work by Ryszard GoÅ‚dyn and associates. The authors have included references to their work on phosphorus internal loading and sustainable lake restoration as follows: ‘…The question of how long it would take to deplete the P pool in lake-bed sediments needs to be addressed to determine sustainable restoration [61-63]…’
Page 20, Lines 638-646: The authors have included the new Ryszard GoÅ‚dyn and associates’ references [54-56] in the revised References section as follows:
- GoÅ‚dyn, R.; PodsiadÅ‚owski, S.; Dondajewska, R.; Kozak, A. The sustainable restoration of lakes—towards the challenges of the Water Framework Directive. Ecohydrol. Hydrobiol. 2014, 14, 68-74. http://dx.doi.org/10.1016/j.ecohyd.2013.12.001
- Dondajewska, R.; Kozak, A.; Kowalczewska-Madura, K.; Budzyńska, A.; Gołdyn, R.; Podsiadłowski, S.; Tomkowiak, A. The response of a shallow hypertrophic lake to innovative restoration measures - Uzarzewskie Lake case study. Ecol. Eng. 2018, 121, 72-82. https://doi.org/10.1016/j.ecoleng.2017.07.010
- Kowalczewska-Madura, K.; Dondajewska, R.; Gołdyn, R.; Rosińska, J.; Podsiadłowski, S. Internal phosphorus loading as the response to complete and then limited sustainable restoration of a shallow lake. Ann. Limnol. - Int. J. Lim. 2019, 55, 4. https://doi.org/10.1051/limn/2019003
Special Issues Journal Water (ISSN 2073-4441)
Manuscript ID water-1171380
Title
Long-Term Oxbow Lake Trophic State Under Agricultural Best Management practices
Authors
Richard E. Lizotte * , Lindsey M.W. Yasarer , Ronald L. Bingner , Martin A. Locke , and Scott S. Knight
Please provide a point-by-point response to the reviewer’s comments and either enter it in the box below or upload it as a Word/PDF file. Please write down "Please see the attachment." in the box if you only upload an attachment. An example can be found here.
* Author's Notes to Reviewer
Reviewer 2
Author's Reply to the Review Report in Bold Font (Reviewer 2)
Review Report Form
Open Review
(x) I would not like to sign my review report
( ) I would like to sign my review report
English language and style
( ) Extensive editing of English language and style required
( ) Moderate English changes required
(x) English language and style are fine/minor spell check required
( ) I don't feel qualified to judge about the English language and style
Yes
Can be improved
Must be improved
Not applicable
Does the introduction provide sufficient background and include all relevant references?
(x)
( )
( )
( )
Is the research design appropriate?
(x)
( )
( )
( )
Are the methods adequately described?
(x)
( )
( )
( )
Are the results clearly presented?
(x)
( )
( )
( )
Are the conclusions supported by the results?
(x)
( )
( )
( )
Comments and Suggestions for Authors
The article is very interesting - presents an analysis of the impact of watershed reclamation on the trophic state of a small oxbow lake over many years. I have no objections to the methods and statistical analyses used in the paper.
I am not surprised by the lack of clear ecosystem response (trophic state improvement) of a shallow water body after reducing allochthonous nitrogen and phosphorus loads. Many years of accumulation of agricultural nutrient loads resulted in the formation of fertile sediments. Lack of thermal stratification and mixing of waters causes resuspension of nitrogen and phosphorus from the sediments and trigger primary production. Such water bodies keep high trophic state long time after the allochthonous nitrogen and phosphorus loads have been reduced. It seems that with such a large internal supply, the only solution is to control not the amount of primary production but rather the type of producers - the transition from a state where algae dominates to a state where primary production run by macrophytes. Many literature data and my observations indicate that it can be done by top-down regulation (strengthening the population of predatory fish, or even temporarily removing fish from the ecosystem).
Information on crop fertilization is worth adding.
Page 4, Lines 131-136: The authors have included crop fertilization information in the text of the manuscript and referenced a revised figure 2 to include a new figure 2b that includes fertilizer information as follows:
‘…Similarly, fertilizer application decisions were made by farmers and typically included applications added in spring months (March-June) prior to planting a crop, and was usually knifed in. Nitrogen fertilizer, primarily as urea-ammonium-nitrate, was applied every year except 2008-2010. Phosphorus fertilizer was only occasionally applied with the largest application occurring in 2005. Potassium and sulfur were applied frequently from 1998-2005, but only occasionally thereafter (Figure 2b)…’
Page 4, Lines 137-140: The authors have included crop fertilization rate information for the watershed for the study years as a new figure ‘Figure 2b‘ adjacent to revised figure ‘Figure 2a’ (conservation tillage).
line 92 instead of: “acreage of watershed” give “watershed area”
Page 4, Line 128: Correction made by the authors from ‘acreage of the watershed’ to ‘watershed area’.
Authors can find some information about nutrients resuspension in papers published by GoÅ‚dyn R., e.g “Internal phosphorus loading as the response to complete and then limited sustainable restoration of a shallow lake”
Page 16, Lines 421-422: The authors thank the reviewer for bringing to their attention the work by Ryszard GoÅ‚dyn and associates. The authors have included references to their work on phosphorus internal loading and sustainable lake restoration as follows: ‘…The question of how long it would take to deplete the P pool in lake-bed sediments needs to be addressed to determine sustainable restoration [61-63]…’
Page 20, Lines 638-646: The authors have included the new Ryszard GoÅ‚dyn and associates’ references [54-56] in the revised References section as follows:
- GoÅ‚dyn, R.; PodsiadÅ‚owski, S.; Dondajewska, R.; Kozak, A. The sustainable restoration of lakes—towards the challenges of the Water Framework Directive. Ecohydrol. Hydrobiol. 2014, 14, 68-74. http://dx.doi.org/10.1016/j.ecohyd.2013.12.001
- Dondajewska, R.; Kozak, A.; Kowalczewska-Madura, K.; Budzyńska, A.; Gołdyn, R.; Podsiadłowski, S.; Tomkowiak, A. The response of a shallow hypertrophic lake to innovative restoration measures - Uzarzewskie Lake case study. Ecol. Eng. 2018, 121, 72-82. https://doi.org/10.1016/j.ecoleng.2017.07.010
- Kowalczewska-Madura, K.; Dondajewska, R.; Gołdyn, R.; Rosińska, J.; Podsiadłowski, S. Internal phosphorus loading as the response to complete and then limited sustainable restoration of a shallow lake. Ann. Limnol. - Int. J. Lim. 2019, 55, 4. https://doi.org/10.1051/limn/2019003
Round 2
Reviewer 1 Report
No more changes are required.